# The influence of erotic camsites on improving men's body comfort: A qualitative analysis of mechanisms

Amanda N. Gesselman[1,2]*, Ellen M. Kaufman[1], Jessica T. Campbell[1,3], Margaret Bennett-Brown[1,4], Melissa Blundell Osorio[1,5], Camden Smith[4], Malia Piazza[1,6], Zoe Moscovici[1,5]

**1** The Kinsey Institute, Indiana University, Bloomington, Indiana, United State of America, **2** School of Nursing, Indiana University, Indianapolis, Indiana, United States of America, **3** Center for Evaluation, Policy, and Research, Indiana University, Bloomington, Indiana, United States of America, **4** Department of Communication Studies, Texas Tech University, Lubbock, Texas, United States of America, **5** Department of Gender Studies, Indiana University, Bloomington, Indiana, United States of America, **6** Department of Anthropology, Indiana University, Bloomington, Indiana, United States of America

* agesselm@indiana.edu

## Abstract

A significant portion of U.S. men experience body dissatisfaction, which can be harmful and limiting to their overall quality of life and well-being. Positive interactions, especially those occurring in the context of sexual behavior, have the potential to enhance men's body self-perceptions. In this study, we explored the impact of engagement with erotic camsites on men's comfort with their own bodies via web-based survey. A sample of 5,828 predominantly heterosexual, cisgender men recruited from LiveJasmin.com reported their demographics, camsite behaviors, and whether camsite use increased their body comfort. Our findings revealed that 19% of participants ($n$ = 1,088) reported increased body comfort. We thematically coded qualitative responses to identify mechanisms for increased comfort. Key mechanisms included receiving positive feedback from models (i.e., performers), engaging in self-exposure on video, discussing body and sexual preferences with models, and experiencing perspective shifts. These results provide preliminary evidence that camsites are interactive environments in which feedback and self-expression might positively influence body comfort. While these findings suggest potential benefits, they also raise questions about the broader implications of such digital interactions for body image, highlighting the need for further research to understand the complexities of these technologically-mediated spaces.

## Introduction

Self-perceptions about one's body are critical to mental and physical well-being, significantly contributing to self-esteem, psychological resilience, and overall quality of life [1–3]. While women often report heightened struggles with body image [4], body dissatisfaction is also highly prevalent for men [5–6]. A recent national study reported that 27% to 36% of U.S. men are dissatisfied with at least one aspect of their body [7]. This dissatisfaction highlights

**Data availability statement:** All data files and codebooks are available from the OSF database at https://osf.io/cwtgy/.

**Funding:** This project was funded by Byborg Enterprises, owner of the website LiveJasmin. com. The funder was not involved in the design of the survey from which the current manuscript draws data, and was not involved in the analyses, writing, or decision to publish the current manuscript.

**Competing interests:** The authors have declared that no competing interests exist.

the need to explore how men perceive their bodies, especially factors that positively influence these perceptions. This study examines how men's engagement with socially-interactive sexual technology—specifically camsites—positively impacts their body perceptions. We investigate how engaging with these digital sexual spaces affects men's comfort with their bodies and explore the mechanisms behind these changes.

## Men's body image

Understanding how men perceive their bodies is crucial for comprehending their overall body image. Body image is a multidimensional construct encompassing individuals' feelings and attitudes towards their own body and physical appearance, including body satisfaction, dissatisfaction, and comfort [8–12]. Contemporary western culture holds demonstrably negative body image, a pattern termed "normative discontent" [5–6]. Factors influencing body image include societal ideals, media portrayals of masculinity, cultural norms, peers and romantic partners [13–15]. Reflecting societal pressures to embody a certain masculine ideal, men's concerns often revolve around muscle distribution, height, and penis size [3,14,16]. Recent studies report that approximately 30% of U.S. men have negative body perceptions, including dissatisfaction with some aspect of their bodies [7,17]. Thus, a significant portion of U.S. men currently have or will develop negative perceptions of their bodies, and these negative perceptions can be harmful and limiting. Positive perceptions, like higher body satisfaction, are linked to higher self-esteem, better quality of life, and reduced risk of depression, eating disorders, and steroid use [14,18,19]. Given the impact of negative body perceptions on well-being, identifying factors that influence body image is crucial.

## Contributors to men's body image

**Interpersonal relationships.** Interpersonal relationships play a pivotal role in shaping men's body image [20–24]. Romantic relationship partners are often the most influential peers for adults [25–26] and can significantly impact body perceptions [27–28]. For example, positive experiences with partners are linked to better body image for men [28–29], while appearance-based criticism from a partner increases men's body dissatisfaction and disordered eating behaviors [30–31]. Men's feelings about their body appear to be linked to sexual partner engagement in particular [29]. Men are more satisfied with their bodies when engaging in frequent sexual activity and derive body satisfaction from a sexual partner's approval [32]. Additionally, some men engage in sexting to reinforce body satisfaction [33–34]. These findings suggest that sexual behavior may positively affect body perceptions for some men.

**Pornography use.** Media effects on body perceptions have long interested researchers, especially the effects of sexually explicit media like online pornography ('*pornography*' hereafter) [4,35]. In the U.S., 46% of men view pornography at least weekly [36]. Pornography consumption is significant in studying men's body perceptions as it frequently exposes viewers to images of naked men, leading to social comparison and body dissatisfaction. Studies show that greater pornography consumption correlates with the internalization of muscular body ideals, dissatisfaction with penis size, cognitive distractions about body image and sexual performance, and lower overall body appreciation and satisfaction [37–40].

Newer forms of technology-mediated sexual engagement, like erotic webcam sites ('*camsites*' hereafter), add nuance to this link. Millions of people use camsites monthly, with over 240,000 users online at any moment [41–43]. Data collected in 2019 indicates that approximately 18% of U.S. adults have visited a camsite [44]. Research has thus far focused on camsites' unique impacts on interpersonal communication skills, mental health, mood, and ability to facilitate emotional intimacy [45–48]. This study is the first to explore how camming affects users' body perceptions.

**Camsites.** Erotic webcam modeling platforms, or "camsites," offer an interactive experience where live performers ("models") engage in explicit acts for site credits [49]. Models perform live shows to an audience, broadcasting video while users communicate via text-based chat, making requests through tips or spending credits [50]. Users can also opt for one-on-one private sessions with models (known as "cam2cam"), allowing interaction through text or audio/video connections. These private sessions provide a more personalized experience and are often preferred, but are more expensive than audience-based sessions.

Considering the link between men's body perceptions, sexual behaviors, and consumption of sexually explicit media, camsite use may affect users' body perceptions. However, camming differs from conventional online pornography in ways that may meaningfully impact how users feel about their bodies. Camsites allow users to co-create sexual performances through real-time requests to models via the chat in audience-based sessions or via text or video connections in private sessions [49]. This interactivity can lead to emotional closeness with models, especially for users who engage in repeated sessions with the same model [45,48,49,51]. These sessions often involve high self-disclosure [45,48,49], which may include disclosures and conversations around sensitive topics like body perceptions. This added emotional intimacy component, not available to viewers of online pornography, may positively impact users' body perceptions.

Additionally, in both audience-based and private sessions, models typically perform alone (i.e., not with a partner) [49]. Most camsite users are heterosexual men viewing performances by models of a different gender [47]. While social comparison drives body satisfaction issues in pornography viewers, camsites may limit such comparisons. Instead, the private camsite context can enhance body positivity through repeated self-exposure. Users in more private sessions may become more comfortable with their own bodies and nudity, as many video platforms display a self-image thumbnail throughout the stream [52].

**Study aim.** In this study, we explored how camsite use influences men's body perceptions, specifically examining whether and why men became more comfortable with their bodies due to camsite engagement. Our findings offer a deeper understanding of how men's body perceptions are affected by their sexual behavior, providing pioneering insights into how emerging sexual technologies can enhance men's body perceptions.

## Materials and methods

### Participants

The participants in this study (*N* = 5,828) were predominantly men. As detailed below, participants were recruited from the camsite LiveJasmin.com. See Table 1 for participant demographics and an overview of their camsite behavior.

### Procedure

The current study was approved by the Institutional Review Board at Indiana University on August 2, 2022 (Protocol #16028). Participants were recruited from LiveJasmin.com in 2023, from February 8 to February 14. LiveJasmin is one of the most popular camming sites, consistently attracting substantial web traffic with approximately 259.15 million visits in April 2024 [42]. After logging into their accounts, they were presented with an advertisement for the survey, displayed as a banner at the bottom of their screen. The banner read, "Help science! Take a quick survey about LiveJasmin." The banner linked participants to an online survey hosted on Qualtrics, associated with the first author's university account. To prevent collecting any identifying information, participants did not sign an Informed Consent form. Instead, they received a Study Information Sheet (SIS) that outlined study procedures, risks and benefits.

**Table 1. Participant demographics and camsite behavior.**

|  | % or mean and standard deviation |
|---|---|
| Gender | |
| Man | 98.5% |
| Woman | 0.8% |
| Another gender not listed | 0.7% |
| Sexual orientation | |
| Straight/ heterosexual | 90.6% |
| Gay/ lesbian | 0.7% |
| Bisexual | 4.4% |
| Pansexual | 1.2% |
| Asexual | 0.3% |
| Questioning or unsure | 1.5% |
| Another identity not listed | 1.4% |
| Race and ethnicity | |
| Asian or Pacific Islander | 5.2% |
| Black or African-American | 2.5% |
| Hispanic or Latino | 8.0% |
| Native American or American Indian | 1.2% |
| White | 84.7% |
| Another race or ethnicity not listed | 3.8% |
| Age | $M$ = 41.0 years, $SD$ = 12.1 years |
| Account length on the LiveJasmin camsite | |
| Less than 3 months | 3.2% |
| 3 to 6 months | 3.9% |
| 6 to 11 months | 2.6% |
| 1 year | 6.8% |
| 2 years | 10.8% |
| 3 years | 12.3% |
| 4 years | 8.1% |
| 5 or more years | 52.4% |
| Number of one-on-one sessions with a LiveJasmin model | |
| None | 2.8% |
| 1-2 times | 5.1% |
| 3-5 times | 6.9% |
| 6-10 times | 7.8% |
| 11-20 times | 8.6% |
| More than 20 times | 68.9% |
| Typical monthly spending on LiveJasmin | |
| Less than $100 USD | 40.2% |
| $100 - $250 USD | 30.6% |
| $251 - $500 USD | 15.1% |
| $501 - $1,000 USD | 7.6% |
| $1,001 - $2,500 USD | 3.9% |
| More than $2,500 USD | 2.6% |
| Hours spent on LiveJasmin in the past month | $M$ = 22.9 hours, $SD$ = 45.5 hours |

Participants clicked to indicate their agreement to participate, then completed a brief survey exploring changes in knowledge, attitudes, or behaviors as a function of camsite use. Participation was voluntary, and participants were not compensated. See https://doi.org/10.17605/OSF.IO/CWTGY for a copy of the survey and access to data used in the current study.

Byborg Enterprises—the owner of LiveJasmin.com—funded development of the larger study, but was not involved in the creation of the survey instrument, data analyses, manuscript writing, or decisions around publication. The research team and funders made the a priori decision to promote the survey for one week, recruiting as many participants as possible during that time frame; no a priori power analyses were conducted.

## Measures

**Participant demographics.** Participants reported their age, gender, race and ethnicity, and sexual orientation.

**Camsite behavior.** Participants reported how long they had an account on LiveJasmin (*less than 3 months*; *3–6 months*; *6–11 months*; *1 year*; *2 years*; *3 years*; *4 years*; *5 or more years*), how many hours they spent on LiveJasmin in the last month (open text entry; responses beyond possible hours in a month and outliers beyond +/- 2*SD* coded as missing), how many times they had one-on-one sessions with cam models on LiveJasmin (*never, once or twice, 3-5 times, 6-10 times, 11-20 times, more than 20 times*), and how much money they typically spend per month on LiveJasmin (*less than $100, $100-$250, $251-$500, $501-$1000, $1001-$2500, more than $2500*, and *prefer not to answer*).

**Body comfort.** Participants responded to a survey question assessing whether they had experienced changes because of their camsite use. The question read, "During your time on LiveJasmin, have any of the following occurred?" This was followed by a list of potential changes (e.g., increased knowledge about sex in general, changes in one's sexual preferences) that allowed participants to select all that applied. This study specifically examines responses to the statement, "I've become more comfortable with my body." Those who identified with this change were then directed to a subsequent question: "Since you've been using LiveJasmin, what has helped you become more comfortable with your own body?" Here, participants were given an open text box to elaborate on their experiences.

## Data analysis

We explored descriptive statistics to determine if participants reported an increase in comfort with their bodies, using a binary measure (1 = did become more comfortable, 0 = did not). Participants who reported an increase in comfort with their bodies were provided an open text box to describe what has helped them become more comfortable. To analyze the qualitative responses from that open text box, we employed thematic analysis following Braun and Clarke [53], which allowed us to identify themes or patterns within the data. This inductive approach enabled themes to emerge directly from the data without a pre-established coding framework. Participants entered their responses directly into the survey, preserving their original form for analysis.

We began our thematic analysis by consolidating all responses into a single spreadsheet. To develop an initial codebook, we selected a random 10% of these responses for preliminary analysis. Together, we reviewed this subset to identify and generate codes representing participants' sentiments, forming our first codebook. Independently, we then coded the full set of qualitative responses. Recognizing the need for broader coverage, we expanded and refined our codebook, eliminating redundant codes and grouping the remaining ones into primary themes and subthemes. After two rounds of refinement, our codebook was finalized.

We proceeded to independently recode all responses using the final codebook. With eight coders, our interrater reliability was satisfactory at $k = .80$. For responses where fewer than five of the eight raters assigned the same code ($n = 53$), we reviewed and discussed until reaching consensus. Each response was assigned a single code representing its primary mechanism.

## Results

Of the overall sample ($N = 5,828$), 19% ($n = 1,088$) reported becoming more comfortable with their body during their time on the LiveJasmin website.

## Qualitative responses

Within the group of 1,088 participants, 477 (44%) provided a response about what has helped them feel more comfortable with their body since they began using LiveJasmin. We identified seven overarching themes within those responses, along with additional 'miscellaneous' and 'not applicable' categories. The themes reflecting reasons/mechanisms for increased body comfort were: (1) response from models; (2) reflection/ perspective shift; (3) physical self-exposure; (4) cam2cam technology; (5) talking to models; (6) body positivity; and (7) altering one's body or appearance; along with (8) miscellaneous and (9) not applicable. Themes are described in detail and illustrated with quotes below; see Table 2 for an overview and definition of themes along with representative quotes. To maintain the integrity of participants' responses, we chose to present quotes unedited.

**Response from models.**  Among the mechanisms cited for increased body comfort, one in four participants (24.5%, $n = 117$) highlighted the impact of models' behavior. Many participants mentioned receiving positive verbal feedback or affirmation from the models, particularly in the form of compliments. These compliments, such as "Having her tell me I am so handsome and how my cock is so big and hard," were noted to improve their body perceptions. Even though clients acknowledged that part of the models' job is to provide compliments—for instance, "Compliments about my appearance (even my penis, even though they're probably just saying it because they have to!) – the women are often (it seems) pleasantly surprised when they see me on cam"—this reality did not diminish the positive impact of such affirmations, especially when received consistently. One participant elaborated:

> "I got bullied a lot when I was young and didn't have a very positive view of myself sexually. I know models are paid to be nice, I'm not an idiot, but when you get consistently complimented on the same things trends develop and it becomes easier to accept those compliments as true. Combine that with the sexual attention and it's been a big boost to my self confidence that has lead me to a much higher level of satisfaction in real life/offline."

Positive feedback from models was also noted to be non-verbal. Participants highlighted that models' emotional reactions—both verbal and non-verbal—upon seeing their bodies bolstered their body perceptions. For example, one participant said, "Many girls seem to really enjoy seeing my dick and that gives me pleasure and comfort." Similarly, another said, "Feedback from models, seeing their reactions to me". The pleasure models seemed to derive from viewing a client's body further amplified this effect, as one participant explained:

> "Being able to clearly see how women are impressed when they see my body. As an active young man, I frequently suffer from body dysmorphia, and so it can be hard to realise how far I've come in my own journey; seeing how women have enjoyed looking at my body has helped me significantly."

**Table 2. Qualitative themes and representative quotes.**

| Theme | Definition | Example quotes |
|---|---|---|
| Response from models (25%; *n* = 117) | Participants have become more comfortable with their bodies as a result of receiving positive affirmation—or lack of a negative reaction—from the models about themselves or their bodies. The models' response could be verbal or non-verbal. | "My body has been complimented." "I've gotten confirmation from women that I'm also attractive to them & not to feel embarrassed." "I have never been judged by any of the models (although I am paying for the time so that could be part of it) but it does help with self esteem" |
| Reflection/perspective shift (15%; *n* = 74) | Participants became more comfortable with their bodies as a result of an epiphany, a personal reflection or introspection about bodies, or a shift in their own perspectives about bodies. | "I don't have to be perfect" "No bodies are perfect" "realizing we're all the same" |
| Physical self-exposure (10%; *n* = 48) | Participants have become more comfortable with their bodies as a result of exposing part of all of their bodies to models on the site. | "Showing my body on webcam" "Showing it even when I got more kilos on" "Displaying it in pvt has made me more confident. Women are more appreciative of how I look if I am okay with the way I look" |
| Cam 2 cam (10%; *n* = 47) | Participants have become more comfortable with their bodies as a result of the camming feature on the website or the experience of being on camera/ video. | "c2c" "Cam 2 Cam" "going on camera" |
| Talking to models (4%; *n* = 21) | Participants have become more comfortable with their bodies as a result of having body-related or sexual fantasy discussions with models. This differs from 'Response from models,' which is centered around models' reactions or compliments, rather than conversation or discussion about bodies. | "Talk about my body with women" "speaking to models, hearing thier perspectives" "talking openly about sex has helped me understand that I don't need a perfect body" |
| Body positivity (4%; *n* = 17) | Participants have become more comfortable with their bodies as a result of exposure to other people's bodies that go against societal norms for beauty or that have expanded their own beauty standards. | "Being with women who are not Barbie doll perfect has made me more accepting of my body" "Exposure to average females" "Seeing no shame in others" |
| Altering body or appearance (2%; *n* = 9) | Participants have become more comfortable with their bodies as a result of changing their body or how they look—either on camera or outside of the camming context. | "My body when I crossdress" "I LOST WEIGHT, AND MY BODY IS HOTTER LOOKING THAN 4 YEARS AGO" "Losing weight at the same time and kindness of the models" |
| Miscellaneous (18%; *n* = 87) | These responses did not give enough context or information to understand how participants' body comfort increased. | "That I have a nice cock" "Trying new things" "Knowing that there are risks, and being willing to be adventurous a little with my own body hasn't been a bad thing." |
| Not applicable (12%; *n* = 57) | These responses were unrelated to the question, avoided answering the question, or were seemingly meaningless. | "Good" "not much" "Sorry can't explain it's personal" |

Additionally, models' behaviors such as mutual masturbation further validated the positive feedback, suggesting genuine arousal by the client's body.

The absence of negative feedback or judgment also played a role in shaping clients' body perceptions. This was evident in both verbal and non-verbal feedback indicating that a client's body did not need to conform to conventional standards to be considered desirable. As one participant noted, "The performers helped me to understand that I don't have to look like a porn star to not be disgusting. That even though I might not have the most beautiful figure I can still be sexy."

Some clients suggested that models' deliberate de-emphasis on physical appearance, and their attention to other personal qualities, contributed to increased body comfort. One client explained, "The models don't always value the body the most, but rather the person. The size is enough and a huge size doesn't mean better sex. It's something that's made me more comfortable." Another client emphasized that being more than their physical appearance reduced their body insecurities: "I don't consider myself especially attractive due to being overweight

and below average in genitalia size. But the emotional connections I have built with some models makes me believe I am desirable." Ultimately, various forms of feedback from models contributed to clients feeling more comfortable with their bodies. As one participant summarized, "I have done cam to cam, and they never judge and compliment your body, so in a way that makes me feel more comfortable, I have never done something like this in my life."

**Talking to models.** Although thematically similar to responses coded as 'Response from models,' 4.4% (*n* = 21) of participants specifically highlighted how having body-related discussions with models—rather than receiving compliments or observing the models' behavior with regards to the participants' bodies, as in 'Response from models'—helped them experience greater comfort with their bodies. Responses in this category mentioned participants discussing their own body with the models or discussing bodies in general. For instance, one participant said that "seen and talking to models about bodies and stereotypes" helped to increase body comfort. Other participants indicated that being able to discuss their sexual fantasies and preferences with the models also affirmed their comfort in their bodies. For example, one said, "I have become more comfortable with my body as I have shared my sexual fantasies with partners." These responses emphasize that one of the critical benefits of interacting with cam models is the opportunity for clients to articulate their feelings or perspectives about their bodies or sexual experiences.

**Reflection/perspective shift.** Many respondents reported experiencing increased body comfort due to reflection or a shift in perspective. More than 15% (*n* = 74) of responses indicated that participants had become more comfortable with their bodies following self-reflection or introspection. Some participants noted a change in comfort without being able to pinpoint the exact mechanism. For example, one said, "i dont know why but i feel more self assured." Additionally, several participants expressed that they had become more accepting of themselves, particularly regarding their bodies; for example, "my size and body type is okay" and "even though i'm not in the bhest shape i'm more accepting of it."

Participants also reported a positive shift in perspective around their genitals, leading to improved feelings towards their penis. One participant reported, "I have become less insecure about my penis size." Relatedly, one said, "i feel free with been nude here and feel ok with the size of my dick.i found average is ok with most woman and breaks the big lie that woman like bigger dicks." In some cases, participants became more accepting of changes their bodies had undergone. For example, one respondent shared,"i dont have balls due to cancer and i was self-conscious about that but not so much now."

Additionally, several participants reported that their shift in perspective led them to let go of the idea of a perfect body. As one participant stated, "I realize body perfection is mythical, and not what necessarily makes you attractive." Others reported feeling less shame and self-consciousness around their body as a result of using the camsite, with one participant stating, "I'm more willing to show my body without the fear of judgement." Finally, some participants reported experiencing greater comfort, confidence, and ease with their physical form. For example, one participant stated, "That their are all kinds of people that like all types of men which has led to me being more confident."

**Cam 2 cam and physical self-exposure.** Respondents identified one feature of the LiveJasmin site as a key factor in improving body comfort: cam2cam (or c2c), which enables private one-on-one interactions between a model and a client. Nearly 10% (*n* = 47) of participants indicated that the c2c feature enhanced their body perceptions, including one who said, "When I go private I use cam 2 cam and it definitely builds confidence." The computer-mediated nature of camming—being an online sexual encounter—allowed clients to communicate with models without being physically present. As one respondent noted, "The internet separation makes me much less selfconscious." While only one respondent

mentioned distance specifically, other responses highlighted communication via cam2cam as beneficial. For example, one participant stated, "The video cams allowing me to chat with the models helped me become less shy about my body."

The c2c feature also includes the option for clients to turn on their own camera, introducing the theme of Physical Self-Exposure (10.1%, $n = 48$). This theme emphasizes that exposing oneself (partially or entirely) on camera improved self-perceptions of body comfort. This theme highlights the act of self-exposure as the mechanism improving body comfort, with participants reporting, "Getting naked online to total strangers kind of gave me confidence in real life," and "The site has a feature to turn in your own camera and show to your partner so showing my naked body to a stranger has helped."

Together, the c2c feature and the ability to expose oneself enabled clients to interact with models more intimately while embracing bodily insecurities. Several participants noted that this intimacy through c2c and physical self-exposure helped them overcome negative self-perceptions, including weight, age, and genitalia size. For example, "I was always self concious about my dick size and my body but having multiple women see them has made me feel more confident in myself." Another respondent explained:

> "I am more proud of my aging but very firm body, and that i like to show off my body, and certain areas a lot, to any model in private, or in my own personal life. I have learned that i like to show off to a willing model that wiil do the same to me and with me. Yes i have love to strip to nude, and let them see me, and enjoy it."

In this response, the participant noted that these features offer a way to embrace bodies at any age, with positive feedback and reciprocity from the model being important additional benefits of the computer-mediated environment.

Several others discussed model responses and positive feedback within private cam sessions. As one participant stated, "I am less shy about my body from using cam to cam with the models. this also comes from sharing my cam and receiving positive comments that seem honest." Another participant echoed these sentiments, stating:

> "Cam2Cam experiences, where gorgeous sexy models will ask to see my naked body, my genitals, and to see me engaged in self-pleasuring activities. This tends to make me more comfortable with my body and with sharing it."

These examples point to the intimate and private environment of camming, where clients' self and physical exposures are met with acceptance and positive affirmation by models, fostering improved bodily comfort and perceptions.

**Body positivity.** Some respondents reported that their increased body comfort was due to viewing a variety of body types and sizes on the platform, or by observing body-positive attitudes where an imperfect body was not a barrier to participation. Nearly 4% ($n = 17$) of respondents indicated that exposure to other people's bodies, which did not conform to societal definitions of perfection, improved or expanded their own sense of beauty standards and bodily acceptance. Some participants noted that simply seeing imperfect bodies on the platform made them feel better about their own bodies; as one participant noted, "spending time with experienced cam models will show you that no one is physically perfect. even the models you think are 10/10 will show their flaws to the camera eventually." Others highlighted the value of seeing a diversity of body types and sizes, stating that "Seeing so much variety" or "seeing all shapes" improved their own body comfort.

In some cases, the impact on participants' body comfort came from observing others with imperfect bodies being comfortable with themselves, demonstrating that not having a perfect body was not a barrier to physical comfort. For example, one participant stated that "Seeing people comfortable showing their body regardless of their own appearance" improved their body comfort. For one participant, it was specifically observing a variety of body sizes in sexual settings that led to increased comfort with their own body. This participant stated, "seeing people of all body sizes in sexy situations".

**Altering body or appearance.** A select 1.9% of responses noted experiences in which altering their self-presentation—either within or outside of the context of camming—aided in increased body comfort. These responses ($n = 9$) were coded as Altering Body or Appearance and listed the distinct changes respondents made to achieve a more ideal body or appearance. Three respondents described making changes to their clothing. For example, one respondent said they "love to get dressed in latex." The other two respondents described dressing in a way that is associated with a gender different than their identification, such as "dressing in female clothes." While these responses do not explicitly state that these changes were done or shown on the LiveJasmin site, they indicate that the camming environment can foster a space where clients can comfortably and privately communicate aspects of themselves, perhaps in ways they cannot achieve offline.

The remaining responses within this theme pertained to altering one's body or appearance through working out and/or weight loss, as illustrated by this participant response: "I've become way more comfortable due to trying to look better and getting in shape." Three respondents also mentioned that models' positive feedback on these changes in participants' appearance improved their body comfort and body perceptions. For example, one participant said, "i know what part of my body is getting better and better while i'm working out and the comment on it by models makes me more confident." These responses reaffirm the theme of 'Response from Models' as a prominent mechanism of the camming environment improving body comfort. In particular, models validated the alterations clients made to their body and/or appearance, aiding in the client's positive body perceptions.

**Miscellaneous/not applicable.** Finally, 18.2% of responses ($n = 87$) were coded as Miscellaneous—relevant to the question, but lacking the necessary information or context required for the research team to fully parse the participants' meaning—with an additional 11.9% of responses ($n = 57$) coded as inapplicable to the question. Most miscellaneous responses were those where the participant expressed some sentiment about their body or body comfort but were not specific about the mechanism through which this change was achieved. For example, one participant noted, "i feel better with my body," but did not specify how this improvement was achieved. A number of these responses specifically discussed their genitalia without context, making it difficult for the research team to identify how (e.g., via a verbal response from a model or physical exposure of these body parts) the clients' professed confidence in their genitalia was connected to their body comfort. Some examples of these responses include, "My cock size" and "my dick." Similarly, other participants mentioned some sexual behavior or activity as increasing their body comfort, but these responses also lacked the necessary context to provide clarity about the participants' experience and why those experiences increased body comfort. For example, one participant stated, "exploring anal sex."

A number of responses in this category also seemed to misunderstand the question, seemingly providing the outcome of their change in body comfort rather than explaining the underlying mechanism for this change. For example, "Being naked with strangers is now is easier." Because these responses were relevant to the research question, they were coded as "miscellaneous" rather than "not applicable." By contrast, many of the 'not applicable'

responses were short, often one-word answers—such as "yes" or "nothing"—that did not provide adequate insight into the participants' sentiment nor their understanding of the question.

## Discussion

The present study examined the impact of camsite use on men's body perceptions. In particular, we focused on how these digital sexual spaces could improve men's comfort with their bodies through a combination of engagement with unique technological features and interactions with cam models. This study fills a gap in existing research by providing novel insights into how more interactive forms of technology-mediated sexual engagement can enhance men's body perceptions.

Our findings revealed that 19% of camsite users reported increased comfort with their bodies—a substantial proportion, indicating that technology-mediated, sexually expressive environments could significantly promote positive body perceptions. This has important implications for interventions aimed at improving body image and reducing related concerns, particularly as virtual interactions evolve. Our qualitative analysis illuminated the mechanisms behind these changes, identifying themes such as responses from models, self-reflection, and physical self-exposure as contributing factors to enhanced body comfort. These findings highlight the complex interplay between technological features and interpersonal dynamics in influencing men's body perceptions.

The most prominent theme was 'Response from models', wherein 25% of participants cited receiving compliments and affirmations from models as significant in improving their body comfort. This suggests that, within this context, camsite models can function as influential actors who provide crucial positive reinforcement to clients. While some participants recognized that giving compliments may be part of the models' job, this awareness did not diminish the positive impact. The consistency of compliments over time helped some participants internalize these affirmations as true, boosting self-confidence and fostering a more positive self-view of themselves. Additionally, models' non-verbal feedback played a significant role in enhancing body comfort. Participants noted that models' reactions, including positive non-verbal cues upon seeing the participant's body, contributed to their increased body comfort. This feedback, coupled with the absence of negative feedback or judgment, helped participants feel more desirable and less focused on conventional body or beauty standards.

Although this only manifested in 4.4% of responses, the theme 'Talking to models' reflects distinct mechanisms for improving body comfort. This theme emphasized communication beyond compliments and reactions—as in the 'Response from models' category—including thorough discussions about bodies, sexual fantasies, and preferences. Some participants specifically valued openly discussing their bodies with models, while others highlighted sharing and exploring sexual fantasies as affirming their body comfort. This dialogue with models, whether explicitly about body image or general desires, was seen by some participants as crucial for fostering a more accepting and positive self-view. Articulating feelings about their body or sexual experiences in a supportive environment was underscored as a key benefit of these interactions.

Another key mechanism identified was self-reflection and shifts in perspective. Approximately 16% of participants explained that their time on LiveJasmin led to significant introspection and cognitive shifts, resulting in increased body comfort. Some participants provided discrete explanations, while others described large-scale changes in understanding and accepting their bodies. Several participants noted a shift in perspective specifically regarding their genitalia, aligning with existing literature on concerns about penis size among men [14,16]. Although this literature often focuses on sexual minority men, our predominantly heterosexual sample also reported changes, indicating its broader relevance. Participants

mentioned becoming less insecure about their penis size and feeling freer and more accepting of their bodies overall. These reflections highlight how the camsite environment may foster self-acceptance and reduce body-related anxieties.

Physical self-exposure and cam2cam features emerged as notable mechanisms for increased body comfort, with 20% of participants' responses falling into these themes. The act of exposing oneself on camera helped participants become more comfortable with their bodies, aligning with research on the benefits of self-exposure and body positivity [54]. The cam2cam feature allows participants to show their bodies while maintaining distance and control, enhancing comfort. This setup lets users decide how much to expose, when to turn the camera on or off, and when to end the session, providing significant agency. The combination of distance and familiar, comfortable surroundings creates a sense of privacy and anonymity [49], contributing to participants feeling safe in revealing themselves. This context allows body exposure without the fear of judgment from familiar individuals, fostering a secure environment for self-exploration. Additionally, the presence of the model adds a unique layer; models often initiate vulnerability by exposing themselves first, inviting users to do the same. This preemptive vulnerability by the model means users do not have to make the first move or break the ice with nudity, potentially reducing anxiety and making it easier for them to engage in self-exposure. The structured and controlled environment of cam2cam interactions may thus foster a supportive space for body acceptance and comfort.

Noticing the diversity of models' bodies was central to nearly 4% of participants' responses about their change in body comfort. Respondents noted that body diversity on the camsite included variations in shape, weight, age, race, and physical features, with exposure to this diversity leading to greater body comfort. Societal norms often limit perceptions of attractiveness, but seeing diverse bodies on camsites may challenge these norms. Participants noted that viewing a variety of body types—especially in a sexual context—helped them realize that having an imperfect body did not prevent models from engaging in sexual exploration and activity. Overall, exposure to diverse bodies and the confidence of models in showing their bodies helped this group of participants—most of whom were men—feel more comfortable with their own bodies.

Finally, a small subset of respondents (1.9%) reported that their experiences on the camsite encouraged them to alter their bodies or change their appearance, increasing their body comfort. Some participants' reported alterations reinforced masculine body norms, including increased muscle tone. Others reported exploring novel forms of self-expression, such as dressing in latex clothing. Although it is unclear how the camming environment prompted these participants to alter their appearance, their responses highlight the dual pressures of conforming to societal norms and the desire for authentic self-expression.

## Limitations

While this study offers valuable insights into how engagement with camsites impacts men's body comfort, several limitations should be noted. First, the sample consisted near-entirely of men who identified as heterosexual and cisgender, limiting the generalizability of the results to populations of men identifying as a sexual minority, as transgender, or as gender non-conforming. Additionally, we asked participants about body comfort rather than body image, which, though similar, are not directly interchangeable. The lack of baseline body comfort data before site usage limits our ability to quantify changes over time. Information on what triggers perspective shifts or discussions about participants' bodies was also not collected. Future studies should gather details such as session frequency before body discussions. Since clients are not required to go on camera, some may never show or discuss their bodies with models, affecting the findings. Future research should examine whether clients go on camera when investigating body comfort or body image.

It should also be noted that our findings reflect the experiences of a relatively small proportion (19%) of the overall sample, and should not be considered generalizable to the general population or even the majority of camsite users. Most participants (81%) in the overall sample, for example, did not report improved body comfort. It is not a requirement for clients to appear on camera or discuss their bodies with models, so for many clients, one's own body comfort may not be deemed relevant during their online sessions. A more critical challenge to interpreting this finding is the phrasing of our measure; because we asked participants if they experienced "more" as opposed to a "change" in body comfort, it is not possible to parse if the majority of participants in the study experienced a decrease in body comfort as opposed to no change at all. This distinction would offer valuable insight for researchers in this area, particularly if findings suggest that the majority of cam site users are experiencing a negative change in body comfort as a result of time spent on the site. These results would run counter to previous research, however, which suggest that negative outcomes (e.g., feeling worse about their sex lives, negative emotional consequences) from camming use are limited to a minority of participants [47]. Nevertheless, we might expect that, consistent with media effects research that links pornography consumption with increased negative body image, some participants may experience decreased body comfort while using the site. Future research should aim to provide participants with a more neutrally-phrased opportunity for sharing their experiences with body comfort to uncover these nuances.

Finally, no data was collected from models. Future studies could explore models' body comfort and the impact of client interactions. Addressing these limitations will enhance understanding of how engagement with camsites impacts men's body comfort.

## Conclusion

In conclusion, this study sheds light on the potentially positive impact of engaging with camsites on men's body comfort, demonstrating that nearly one in five users reported increased body comfort as a result of their camsite interactions. Mechanisms contributing to this improvement including affirmations from models, self-exposure on camera, and perspective shifts facilitated by the camsite environment. These findings suggest that the interactive and sexually expressive nature of camsites provides a unique context for men to explore and accept their bodies. While the transactional nature of these interactions may limit the internalization of positive feedback to some extent, the immediate benefits appear evident. This study underscores the potential of camsites and other forms of interactive sexual technologies to serve as a supportive space for body acceptance, particularly for men who might feel more comfortable discussing their bodies in these settings. However, further research is needed to explore the broader implications for body image and to understand how these dynamics play out in more diverse populations. Addressing the identified limitations, such as the need for baseline data and insights from models, will enhance our understanding of how digital sexual platforms can contribute to positive body perceptions and well-being.

## Author contributions

**Conceptualization:** Amanda N. Gesselman, Ellen M. Kaufman, Jessica T. Campbell, Margaret Bennett-Brown.

**Data curation:** Amanda N. Gesselman, Ellen M. Kaufman, Jessica T. Campbell, Margaret Bennett-Brown.

**Formal analysis:** Amanda N. Gesselman, Ellen M. Kaufman, Jessica T. Campbell, Margaret Bennett-Brown, Melissa Blundell Osorio, Camden Smith, Malia Piazza, Zoe Moscovici.

**Funding acquisition:** Amanda N. Gesselman, Ellen M. Kaufman.

**Investigation:** Amanda N. Gesselman, Ellen M. Kaufman, Jessica T. Campbell, Margaret Bennett-Brown, Melissa Blundell Osorio, Camden Smith, Malia Piazza, Zoe Moscovici.

**Methodology:** Amanda N. Gesselman, Ellen M. Kaufman, Jessica T. Campbell.

**Project administration:** Amanda N. Gesselman.

**Resources:** Margaret Bennett-Brown.

**Software:** Amanda N. Gesselman.

**Supervision:** Amanda N. Gesselman.

**Validation:** Amanda N. Gesselman, Ellen M. Kaufman.

**Writing – original draft:** Amanda N. Gesselman, Ellen M. Kaufman, Jessica T. Campbell, Margaret Bennett-Brown, Melissa Blundell Osorio, Camden Smith, Malia Piazza, Zoe Moscovici.

**Writing – review & editing:** Amanda N. Gesselman, Ellen M. Kaufman, Jessica T. Campbell, Margaret Bennett-Brown, Melissa Blundell Osorio, Camden Smith, Malia Piazza, Zoe Moscovici.

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
