## [Decision Letter · Decision Letter 0]

9 Dec 2024

PONE-D-24-45807The Influence of Erotic Camsites on Men’s Body Comfort: A Qualitative Analysis of MechanismsPLOS ONE

Dear Dr. Gesselman,

Thank you for submitting your manuscript to PLOS ONE. After careful consideration, we feel that it has merit but does not fully meet PLOS ONE’s publication criteria as it currently stands. Therefore, we invite you to submit a revised version of the manuscript that addresses the points raised during the review process.

**ACADEMIC EDITOR:  **

Please review the comments and recommendations made by Reviewers 1 and 2, and consider addressing them and making revisions.

We look forward to receiving your revised manuscript.

Kind regards,

Poowin Bunyavejchewin, PhD

Academic Editor

PLOS ONE

Journal Requirements:

“This project was funded by Byborg Enterprises, owner of the website LiveJasmin.com. The funder was not involved in the design of the survey from which the current manuscript draws data, and was not involved in the analyses, writing, or decision to publish the current manuscript.”

Reviewers' comments:

Reviewer's Responses to Questions

**Comments to the Author**

1. Is the manuscript technically sound, and do the data support the conclusions?

Reviewer #1: Yes

Reviewer #2: Partly

2. Has the statistical analysis been performed appropriately and rigorously? 

Reviewer #1: Yes

Reviewer #2: N/A

3. Have the authors made all data underlying the findings in their manuscript fully available?

Reviewer #1: Yes

Reviewer #2: Yes

4. Is the manuscript presented in an intelligible fashion and written in standard English?

Reviewer #1: Yes

Reviewer #2: Yes

5. Review Comments to the Author

Reviewer #1: I commend the authors for conducting an in-depth investigation of an understudied aspect of online sexual activity, and for using qualitative methods to fill in some considerable gaps in our knowledge about camsite use. I believe the authors offer valuable contributions to the literature, but I have two substantial concerns regarding their presentation of their findings. Pending successful resolution of these concerns, I would be excited to see their manuscript published.

On a smaller note, the manuscript title suggests a broader scope of outcomes (e.g., positive versus negative) than the authors actually report on. The title could more clearly indicate that the manuscript is focused specifically on positive outcomes related to body comfort.

My first critique centers on the use of the tripartite model: I think it is inaccurate to characterize camsite performers as peers of their viewers. In fact, unless I’m reading it wrong, the article the authors cite for the tripartite model adds an additional category of romantic partners to its analyses, rather than sorting them into the “peers” category (Tylka, 2011). Conceptually, I imagine the performers are probably closer to potential romantic partners than to friends in the minds of their clients.

So, I am not sure the tripartite model provides theoretical justification for studying how interactions with performers might affect clients. I do not think the authors necessarily need to provide theoretical justification for their approach, but I think the paper is not helped by the current approach of categorizing performers as peers. Perhaps they could describe in greater detail the studies conducted thus far on outcomes associated with visiting camming sites (i.e., citations 49-51 in the manuscript), which seem to provide ample context and motivation for just a study of this kind.

My second concern is that the authors do not contextualize their findings alongside the 81% of participants who did not describe any positive body comfort-related changes related to their use of LiveJasmin.com. Were any questions asked about potentially negative body comfort-related changes? What might the other four in five site users be experiencing? Not one sentence in this manuscript – I sincerely apologize if I missed it – alludes to the possibility that visiting camsites could also negatively affect men’s body comfort.

I want to be clear that I’m not against any of the findings here – it is encouraging and important that a subset of frequent and highly engaged camsite visitors are experiencing apparent increases in body comfort as a result of their activities on the site. But as at most about 4% of the overall sample is reporting even the most common themes listed here, I would ask that the authors acknowledge in the discussion and/or limitations section(s) that (1) the majority of study participants did not report positive changes in body comfort, and the implications thereof; (2) that specific themes apply to a very small proportion of the overall sample; and (3) that it is possible some of the participants excluded from these analyses are experiencing negative impacts as well (see Milrod & Monto, 2023, which the authors cite). Doing so would open up an avenue for speculation on the authors’ part as to why certain men experience these positive changes in body comfort, while others do not, or may experience negative impacts. Understanding why men fall into these different categories could also be a topic of future research. Doing so would also potentially alleviate concerns among eventual readers that an article funded by the website itself – even though the funders were not involved in the study design and manuscript preparation – is not considering the full range of prior research and potential effects of engaging with camsites.

Reviewer #2: The manuscript “The Influence of Erotic Camsites on Men’s Body Comfort: A Qualitative Analysis of Mechanisms” is incredibly interesting due to its novel assessment of men’s perceptions of their bodies related to engagement with campsites. This manuscript has great potential to add to the literature, however, major revisions to the structuring of the manuscript are warranted to improve the clarity and conciseness of the background, methodology, and results primarily. Please see my tailored recommendations below:

Abstract

1. Make clear the data being assessed are responses from an open text box on a survey that is a subsection of the total sample.

2. Comb through and remove unnecessary prepositions or prepositional phrases, the abstract reads chunky and interrupts the flow.

Introduction

1. Remove the last paragraph, as it is redundant and too similar to your abstract.

2. I would suggest chunking the introduction into sections that highlight (1) the impact of men’s self-perception on their health, (2) the impacts of pornography on perceptions of self, (3) a brief explanation of camsites, and (4) introduce the study aim/goal. The authors have all the parts here, cuts need to be made to reduce the length of the introduction. Potentially aim for 3-3.5 pages max,

Methods

1. Description of the tripartite model, adjustments that were made for this study, and potentially a figure to illustrate how the manuscript findings map on to the model.

2. Add a table 1 of sociodemographic variables; this would greatly reduce the confusion I had about who was surveyed and how many people were assessed for this manuscript.

3. Under “Body comfort,” operationalize what ‘various areas’ are or remove as you list the questions after.

4. In the OSF database, I suggest adding in the blank survey that participants took. You can then reference the survey and reduce word count.

5. Under data analysis, make clear that the authors are analyzing an open text box – otherwise great explanation of the methodology used!

Results

1. Potentially add a table 2 that reports themes, definition of themes, and representative quotes. This will remove some quotes from parentheses and condense the manuscript word count and read cleaner.

2. Stay consistent in how you utilize quotes; some are located in parentheses, while others are not and become jarring to read.

3. Concise conceptualization of results is necessary, and part of that can be mitigated by removing a few unnecessary quotes; however, I found the results impactful nonetheless.

Discussion

1. I am interested in knowing more about the different conceptualizations of response form models and talking to models, perhaps add more in the discussion about that. Additionally, more information about how potential interactions between the identified themes may bolster self-perception.

6. PLOS authors have the option to publish the peer review history of their article (what does this mean? ). If published, this will include your full peer review and any attached files.

**Do you want your identity to be public for this peer review?** For information about this choice, including consent withdrawal, please see our Privacy Policy .

Reviewer #1: No

Reviewer #2: No

---

## [Author Response · Author response to Decision Letter 1]

19 Dec 2024

Dear Dr. Bunyavejchewin and PLOS One reviewers,

Thank you for your time and feedback on our manuscript, “The Influence of Erotic Camsites on Men’s Body Comfort: A Qualitative Analysis of Mechanisms.” We gave serious consideration to all critiques, and believe that our revised manuscript is much stronger than our original submission.

Below, we respond to all feedback we received from the editor and two reviewers, and note how we revised the paper to reflect that feedback. In brief, our most substantial changes to the manuscript were:

1. We removed mentions of the tripartite model from the manuscript. This was originally included as a theoretical justification for our study, but feedback from Reviewer 1 pointed out the weaknesses in this model as a foundation for our research. Reviewer 1 noted that the paper did not need a theoretical foundation, and Reviewer 2 noted that the Introduction section of the paper was already much too long. Taken together, we decided to remove the model from the paper entirely, rather than expand upon the model to make a stronger argument.

2. We revised our Results section, reducing the number of participant quotes included to represent each theme and rewriting some sections to ensure no quotations were included in parentheses, in line with Reviewer 2’s feedback. We also included Table 2, which outlines and defines each theme, and provides representative quotes for each.

3. We included a paragraph in the Limitations section of the Discussion that contextualizes our findings within the full sample of participants. In particular, although we found that 19% of participants reported improved body comfort as a result of camsite usage, the remaining 81% of the sample did not report such an outcome. Our original manuscript did not acknowledge the lack of body comfort improvement in that 81% or discuss potential implications for those individuals. Our revised manuscript does so, in line with Reviewer 1’s feedback.

Although these were our most extensive revisions, please note that we revised every section of the manuscript to align with reviewer feedback. We appreciate the opportunity to revise and resubmit our work to PLOS One. Thank you again for your time and consideration. We look forward to hearing from you!

**Feedback from the Editor**

https://journals.plos.org/plosone/s/file?id=wjVg/PLOSOne_formatting_sample_main_body.pdfaand

--We’ve revised our paper so that all section headers, file names, etc. match the PLOS ONE style guide.

“This project was funded by Byborg Enterprises, owner of the website LiveJasmin.com. The funder was not involved in the design of the survey from which the current manuscript draws data, and was not involved in the analyses, writing, or decision to publish the current manuscript.”

--We’ve revised our financial disclosure to state, “This project was funded by Byborg Enterprises, owner of the website LiveJasmin.com. The funders had no role in study design, data collection and analysis, decision to publish, or preparation of the manuscript.” We’ve included this in our cover letter.

**Reviewer 1 feedback**

1. I commend the authors for conducting an in-depth investigation of an understudied aspect of online sexual activity, and for using qualitative methods to fill in some considerable gaps in our knowledge about camsite use. I believe the authors offer valuable contributions to the literature, but I have two substantial concerns regarding their presentation of their findings. Pending successful resolution of these concerns, I would be excited to see their manuscript published.

--We thank Reviewer 1 for their time reviewing our manuscript. We’ve replied to each of your points of feedback below.

2. On a smaller note, the manuscript title suggests a broader scope of outcomes (e.g., positive versus negative) than the authors actually report on. The title could more clearly indicate that the manuscript is focused specifically on positive outcomes related to body comfort.

--We’ve changed our title from “The Influence of Erotic Camsites on Men’s Body Comfort: A Qualitative Analysis of Mechanisms” to “The Influence of Erotic Camsites on Improving Men’s Body Comfort: A Qualitative Analysis of Mechanisms.”

3. My first critique centers on the use of the tripartite model: I think it is inaccurate to characterize camsite performers as peers of their viewers. In fact, unless I’m reading it wrong, the article the authors cite for the tripartite model adds an additional category of romantic partners to its analyses, rather than sorting them into the “peers” category (Tylka, 2011). Conceptually, I imagine the performers are probably closer to potential romantic partners than to friends in the minds of their clients.

So, I am not sure the tripartite model provides theoretical justification for studying how interactions with performers might affect clients. I do not think the authors necessarily need to provide theoretical justification for their approach, but I think the paper is not helped by the current approach of categorizing performers as peers. Perhaps they could describe in greater detail the studies conducted thus far on outcomes associated with visiting camming sites (i.e., citations 49-51 in the manuscript), which seem to provide ample context and motivation for just a study of this kind.

--Thank you for pointing out this weakness in our manuscript. After reading your feedback and re-reading our paper, we’ve decided to remove the tripartite model completely. We agree that the extent to which we’d included this model wasn’t a strong enough link to our study to create a thorough theoretical argument. We opted to remove it from the paper rather than expand on it because Reviewer 2 noted that our manuscript, and particularly our Introduction section, was too long and needed to be reduced by at least one page of text, and because you—Reviewer 1—noted that this theoretical foundation wasn’t necessary to justify our study’s contribution to the literature.

4. My second concern is that the authors do not contextualize their findings alongside the 81% of participants who did not describe any positive body comfort-related changes related to their use of LiveJasmin.com. Were any questions asked about potentially negative body comfort-related changes? What might the other four in five site users be experiencing? Not one sentence in this manuscript – I sincerely apologize if I missed it – alludes to the possibility that visiting camsites could also negatively affect men’s body comfort.

I want to be clear that I’m not against any of the findings here – it is encouraging and important that a subset of frequent and highly engaged camsite visitors are experiencing apparent increases in body comfort as a result of their activities on the site. But as at most about 4% of the overall sample is reporting even the most common themes listed here, I would ask that the authors acknowledge in the discussion and/or limitations section(s) that (1) the majority of study participants did not report positive changes in body comfort, and the implications thereof; (2) that specific themes apply to a very small proportion of the overall sample; and (3) that it is possible some of the participants excluded from these analyses are experiencing negative impacts as well (see Milrod & Monto, 2023, which the authors cite). Doing so would open up an avenue for speculation on the authors’ part as to why certain men experience these positive changes in body comfort, while others do not, or may experience negative impacts. Understanding why men fall into these different categories could also be a topic of future research. Doing so would also potentially alleviate concerns among eventual readers that an article funded by the website itself – even though the funders were not involved in the study design and manuscript preparation – is not considering the full range of prior research and potential effects of engaging with camsites.

--Thank you for this observation and for the many helpful suggestions for acknowledging and expanding upon this oversight. We have added the following section to the Limitations that directly addresses the generalizability of our findings and touches upon each of the points you mention above:

It should also be noted that our findings reflect the experiences of a relatively small proportion (19%) of the overall sample, and should not be considered generalizable to the general population or even the majority of camsite users. Most participants (81%) in the overall sample, for example, did not report improved body comfort. It is not a requirement for clients to appear on camera or discuss their bodies with models, so for many clients, one’s own body comfort may not be deemed relevant during their online sessions. A more critical challenge to interpreting this finding is the phrasing of our measure; because we asked participants if they experienced “more” as opposed to a “change” in body comfort, it is not possible to parse if the majority of participants in the study experienced a decrease in body comfort as opposed to no change at all. This distinction would offer valuable insight for researchers in this area, particularly if findings suggest that the majority of cam site users are experiencing a negative change in body comfort as a result of time spent on the site. These results would run counter to previous research, however, which suggest that negative outcomes (e.g., feeling worse about their sex lives, negative emotional consequences) from camming use are limited to a minority of participants (Milrod & Monto, 2023). Nevertheless, we might expect that, consistent with media effects research that links pornography consumption with increased negative body image, some participants may experience decreased body comfort while using the site. Future research should aim to provide participants with a more neutrally-phrased opportunity for sharing their experiences with body comfort to uncover these nuances.

Thanks again to Reviewer 1 for helping us to strengthen our manuscript.

**Reviewer 2 feedback**

1. The manuscript “The Influence of Erotic Camsites on Men’s Body Comfort: A Qualitative Analysis of Mechanisms” is incredibly interesting due to its novel assessment of men’s perceptions of their bodies related to engagement with campsites. This manuscript has great potential to add to the literature, however, major revisions to the structuring of the manuscript are warranted to improve the clarity and conciseness of the background, methodology, and results primarily. Please see my tailored recommendations below:

--Thank you to Reviewer 2 for their time and effort reviewing our manuscript. We’ve incorporated nearly all of our feedback into our manuscript. We reply to each of your points below.

2. Abstract: Make clear the data being assessed are responses from an open text box on a survey that is a subsection of the total sample. Comb through and remove unnecessary prepositions or prepositional phrases, the abstract reads chunky and interrupts the flow.

--Thank you for this feedback. We have made edits to the abstract to provide further methodological clarity and to improve the flow of the language.

3. Introduction: Remove the last paragraph, as it is redundant and too similar to your abstract.

--We’ve removed the majority of that paragraph from the manuscript so that it is not redundant with the abstract.

4. Introduction: I would suggest chunking the introduction into sections that highlight (1) the impact of men’s self-perception on their health, (2) the impacts of pornography on perceptions of self, (3) a brief explanation of camsites, and (4) introduce the study aim/goal. The authors have all the parts here, cuts need to be made to reduce the length of the introduction. Potentially aim for 3-3.5 pages max,

--Thank you for this suggestion. We have added headers to the Introduction and have also made edits to this section in the interest of length.

5. Methods: Description of the tripartite model, adjustments that were made for this study, and potentially a figure to illustrate how the manuscript findings map on to the model.

--In response to Reviewer 1’s feedback, we’ve decided to remove the tripartite model from the manuscript. See our reply to their 3rd comment above.

6. Methods: Add a table 1 of sociodemographic variables; this would greatly reduce the confusion I had about who was surveyed and how many people were assessed for this manuscript.

--We’ve added this table and removed text explaining participant demographics.

7. Methods: Under “Body comfort,” operationalize what ‘various areas’ are or remove as you list the questions after.

--We’ve removed “various areas”.

8. Methods: In the OSF database, I suggest adding in the blank survey that participants took. You can then reference the survey and reduce word count.

--We’ve added the survey to the OSF database and included a link in the Methods section.

9. Methods: Under data analysis, make clear that the authors are analyzing an open text box – otherwise great explanation of the methodology used!

--Thank you. We’ve edited the Data Analysis section to read, “Participants who reported an increase in comfort with their bodies were provided an open text box to describe what has helped them become more comfortable. To analyze the qualitative responses from that open text box…”

10. Results: Potentially add a table 2 that reports themes, definition of themes, and representative quotes. This will remove some quotes from parentheses and condense the manuscript word count and read cleaner.

--We’ve added this as Table 2.

11. Results: Stay consistent in how you utilize quotes; some are located in parentheses, while others are not and become jarring to read.

--We’ve removed all quotes from inside parentheses and have reduced the number of quotes included in each theme’s section of the Results.

12. Results: Concise conceptualization of results is necessary, and part of that can be mitigated by removing a few unnecessary quotes; however, I found the results impactful nonetheless.

--See our response to #11 above.

13. Discussion: I am interested in knowing more about the different conceptualizations of response form models and talking to models, perhaps add more in the discussion about that. Additionally, more information about how potential interactions between the identified themes may bolster self-perception.

--We’ve edited the manuscript to make clear the differences between ‘Response from Models’ and ‘Talking to Models’ throughout. For example, in the Results section, we wrote:

“Although thematically similar to responses coded as ‘Response from models,’ 4.4% (n = 21) of participants specifically highlighted how having body-related discussions with models—rather than receiving compliments or observing the models’ behavior with regards to the participants’ bodies, as in ‘Response from models’—helped them experience greater comfort with their bodies.”

Additionally, in the Discussion section, we included the following:

“Although this only manifested in 4.4% of responses, the theme ‘Talking to models’ reflects distinct mechanisms for improving body comfort. This theme emphasized communication beyond compliments and reactions—as in the ‘Response from models’ category—including thorough discussions about bodies, sexual fantasies, and preferences.”

Finally, thank you for your thoughtful suggestion rega

---

## [Decision Letter · Decision Letter 1]

22 Jan 2025

The Influence of Erotic Camsites on Improving Men’s Body Comfort: A Qualitative Analysis of Mechanisms

PONE-D-24-45807R1

Dear Dr. Gesselman,

We’re pleased to inform you that your manuscript has been judged scientifically suitable for publication and will be formally accepted for publication once it meets all outstanding technical requirements.

Kind regards,

Poowin Bunyavejchewin, PhD

Academic Editor

PLOS ONE

Additional Editor Comments (optional):

Reviewers' comments:

Reviewer's Responses to Questions

**Comments to the Author**

1. If the authors have adequately addressed your comments raised in a previous round of review and you feel that this manuscript is now acceptable for publication, you may indicate that here to bypass the “Comments to the Author” section, enter your conflict of interest statement in the “Confidential to Editor” section, and submit your "Accept" recommendation.

Reviewer #1: All comments have been addressed

Reviewer #2: All comments have been addressed

2. Is the manuscript technically sound, and do the data support the conclusions?

Reviewer #1: Yes

Reviewer #2: Yes

3. Has the statistical analysis been performed appropriately and rigorously? 

Reviewer #1: Yes

Reviewer #2: N/A

4. Have the authors made all data underlying the findings in their manuscript fully available?

Reviewer #1: Yes

Reviewer #2: Yes

5. Is the manuscript presented in an intelligible fashion and written in standard English?

Reviewer #1: Yes

Reviewer #2: Yes

6. Review Comments to the Author

Reviewer #1: (No Response)

Reviewer #2: The authors have done an excellent job incorporating all the feedback provided; very good job and this is very interesting work!

7. PLOS authors have the option to publish the peer review history of their article (what does this mean? ). If published, this will include your full peer review and any attached files.

**Do you want your identity to be public for this peer review?** For information about this choice, including consent withdrawal, please see our Privacy Policy .

Reviewer #1: No

Reviewer #2: No

---

## [Editor Report · Acceptance letter]

PONE-D-24-45807R1

PLOS ONE

Dear Dr. Gesselman,

I'm pleased to inform you that your manuscript has been deemed suitable for publication in PLOS ONE. Congratulations! Your manuscript is now being handed over to our production team.

Kind regards,

on behalf of

Dr Poowin Bunyavejchewin

Academic Editor

PLOS ONE